# Effects of Dietary Ferulic Acid on Intestinal Health and Ileal Microbiota of Tianfu Broilers Challenged with Lipopolysaccharide

**DOI:** 10.3390/molecules28041720

**Published:** 2023-02-10

**Authors:** Ziting Tang, Gang Shu, Hong Du, Yilei Zheng, Hualin Fu, Wei Zhang, Cheng Lv, Funeng Xu, Haohuan Li, Ping Ouyang, Juchun Lin, Li-Jen Chang, Felix Kwame Amevor, Xiaoling Zhao

**Affiliations:** 1Department of Basic Veterinary Medicine, Sichuan Agricultural University, Chengdu 611130, China; 2College of Animal Science and Technology, Sichuan Agricultural University, Chengdu 611130, China; 3Center for Veterinary Sciences, Zhejiang University, Hangzhou 310030, China; 4Department of Small Animal Clinical Science, Virginia-Maryland College of Veterinary Medicine, Blacksburg, VA 24062, USA; 5Farm Animal Genetic Resources Exploration and Innovation Key Laboratory of Sichuan Province, Sichuan Agricultural University, Chengdu 611130, China

**Keywords:** ferulic acid, lipopolysaccharide, intestine permeability, intestinal barrier, ileal microbiota

## Abstract

Lipopolysaccharide (LPS) has been considered the primary agent to establish animal models of inflammation, immunological stress, and organ injury. Previous studies have demonstrated that LPS impaired gastrointestinal development and disrupted intestinal microbial composition and metabolism. Ferulic acid (FA) isolated from multiple plants exhibits multiple biological activities. This study investigated whether FA ameliorated intestinal function and microflora in LPS-challenged Tianfu broilers. The results showed that LPS challenge impaired intestinal function, as evidenced by decreased antioxidant functions (*p* < 0.05), disrupted morphological structure (*p* < 0.05), and increased intestinal permeability (*p* < 0.05); however, these adverse effects were improved by FA supplementation. Additionally, FA supplementation preserved sIgA levels (*p* < 0.05), increased mRNA expression levels of *CLDN* and *ZO-1* (*p* < 0.05), and enhanced epithelial proliferation (*p* < 0.05) in the ileal mucosa in LPS-challenged chickens. Moreover, FA supplementation rectified the ileal microflora disturbances in the LPS-challenged broilers. The results demonstrate that dietary FA supplementation decreased LPS-induced intestinal damage by enhancing antioxidant capacity and maintaining intestinal integrity. Furthermore, FA supplementation protects intestinal tight junctions (TJs), elevates secretory immunoglobulin A (sIgA) levels, and modulates ileal microflora composition in LPS-challenged broilers.

## 1. Introduction

Broilers are raised under an intensive husbandry environment in commercial poultry farms, which may increase the risk of ingesting various toxins, such as mycotoxin, bacteriotoxin, and zootoxin. Lipopolysaccharide (LPS) is a crucial bacterial endotoxin produced by most gram-negative bacteria [1]. It has been considered the primary agent to establish animal models of inflammation, immunological stress, and organ injury [2,3,4,5,6,7]. Moreover, the accumulation of excessive free radicals [8] induced by LPS damaged the intestinal barrier [9] and decreased the growth performance of broilers [10] by triggering inflammation of the gastrointestinal tract [11], initiating oxidative damage and apoptosis of intestinal epithelial cells [12,13], and disrupting intestinal microbial composition and metabolism [14,15].

A number of studies indicated active plant components enhanced growth performance and combated various diseases, therefore, had great potential for generating revenue in the poultry industry [16,17,18,19]. An active ingredient of phenolic, ferulic acid (FA, 4-hydroxy-3-methoxycinnamic acid), is isolated from multiple plants and has been recognized as an antioxidant. A series of recent studies indicated that FA shows low toxicity [20] and possesses varieties of biological activities, including antibacterial [21], antioxidant [22], anti-inflammatory [23], antithrombotic [24], and anticancer [25] properties. Additionally, it has been reported that FA supplementation improves the immune functions and growth performance in male adult zebrafish by elevating the number of intestinal goblet cells and ameliorating the intestinal microbiota composition [26]. In piglets, dietary FA supplementation exhibited a similar effect by promotion of Claudin-1 (CLDN-1) and Occludin (OLDN) expression, increasing the Firmicutes/Bacteroidetes ratio (F/B), the abundance of the Lachoiraceaea family, and reduction in the abundance of the Prevotellaceae family in the cecum [27]. Another example of this is that FA reduces atherosclerotic injury by regulating lipid metabolism and intestinal microbiota [28]. However, information with regard to the mechanism of action and how FA protects the intestinal health and microbial composition of broilers remains unreported.

On account of the multiple benefits on intestinal FA, this study was carried out to explore whether dietary 100 mg/kg FA supplementation ameliorates the intestinal injury and intestinal microflorae of broilers challenged with LPS.

## 2. Results

### 2.1. Intestinal Morphological Analysis

Figure 1 represents the morphological characteristics of the ileum among groups. The ileal VH and CD in the LPS group were significantly lower (*p* < 0.05) than in the CON group. While the contrary, it increased significantly (*p* < 0.05) with dietary FA supplementation. Birds in the FL group had the highest VH/CD (*p* < 0.05) among the groups. Similarly, LPS challenge significantly decreased (*p* < 0.05) duodenal VH and VH/CD, while significantly increasing (*p* < 0.05) the CD in the duodenum and jejunum, which were modified by FA supplementation (Appendix A).

### 2.2. Intestinal Permeability Biochemical Analysis

Figure 2 shows the parameters indicating intestinal permeability. The levels of DAO and D-LA in the serum were significantly increased (*p* < 0.05) after LPS challenge; however, FA significantly decreased the serum DAO activity and D-LA levels (*p* < 0.05) in chickens challenged with LPS.

### 2.3. Antioxidant Parameters of Intestinal Mucosa

The levels of antioxidants in intestinal mucosa were examined and shown in Figure 3. The LPS group had significantly lower levels of SOD, T-AOC, and GSH in the mucosa than the CON group (*p* < 0.05). Conversely, levels of SOD, T-AOC, and GSH were significantly elevated (*p* < 0.05) in the FL group when compared to the LPS group.

### 2.4. sIgA Content in Ileal Mucosa

Figure 4 shows that ileal sIgA contents in LPS-challenged chickens were the lowest (*p* < 0.05); however, dietary FA supplementation increased (*p* < 0.05) the ileal sIgA contents in the FL group compared to the LPS group.

### 2.5. Life Cycle of the Ileal Epithelium

Figure 5 shows an increased ileal epithelium percentage in the G_0_/G_1_ phase (*p* < 0.05), a reduced percentage of cells in the S and G_2_M phases, and a lower PI index in the LPS group (*p* < 0.05) compared to the CON group. However, more ileal epithelium entered the S and G_2_M phase (*p* < 0.05) in the FL group compared to the LPS group.

### 2.6. Relative mRNA Expressions of Intestinal Tight Junction Proteins

Figure 6 depicts the relative mRNA expressions of intestinal TJs. The *OCLN* expression did not significantly change, but the *CLDN-1* and *ZO-1* decreased significantly (*p* < 0.05) in the LPS group when compared to the CON group. Significant upregulations (*p* < 0.05) of *CLDN-1* and *ZO-1* were noticed in the FL group compared to the LPS group.

### 2.7. Ileal Microbiota Composition

The Venn diagram (Figure 7A) and flower diagram (Figure 7B) showed the unique and shared OTUs of the different microbiome groups in the ileum. The diagrams illustrate that 1049 OTUs (the core) were shared by four groups. In total, 327, 221, 153, and 127 unique OTUs were discovered in the CON, LPS, FA, and FL groups, respectively (Figure 7B). Interestingly, chickens in the LPS and FL groups shared fewer OTUs (1342) than those in the CON and LPS groups (1663). Among these unique OTUs, 175 OTUs were discovered in both the CON and LPS groups; 136 OTUs were discovered in both the CON and FA groups; 133 OTUs were discovered in both the CON and FL groups; 90 OTUs were discovered in both the LPS and FA groups; and 97 OTUs were discovered in both the LPS and FL groups.

The alpha diversity (Figure 8), a parameter that evaluates diversity and uniformity of the bacterial species, of the ileal microbiota was not significantly (*p* > 0.05) different among the four groups. The ileal microbiota was obviously different among the four groups (PC1, 44.96%; PC2, 15.9%) based on the results of principal coordinate analysis (PCoA) with weighted unifrac distances (Figure 9A). Furthermore, the results (Figure 9B) indicated that the FL group had a similar ileal microbiota composition to the CON and FA groups compared to the LPS group.

Figure 10 presents the composition of microbiota in Tianfu broilers. At the phylum level (Figure 10A), the ileal microbiota of the Tianfu broilers mainly was composed of Bacteroiota (44.3%), Firmicutes (40.23%), Euryarchaeota (4.16%), Desulfobacterota (1.41%), unidentified_bacterira (1.42%), and Antinobacteriota (3.15%). After LPS challenge, the ileal microbiota exhibited an increment of Bacteroidota, Desulfobacterota, and Actinobacteriota, but a decrement of Firmicutes and Euryarchaeota in comparison to the CON group. However, the results showed an increment of Firmicutes and Euryarchaeota, and a decrement of Bacteroidota, Desulfobacterota, and Actinobacteriota in the FL group when compared to the LPS group.

At the family level (Figure 10B), Bacteroidaceae (22.96%) was the most abundant family followed by Rikenellaceae (16.94%), Lachnospiraceae (13.63%), Lactobacillaceae (7.56%), Ruminococcaceae (7.05%), and Muribaculaceae (2.45%). LPS reduced the abundance of Lactobacillaceae and Ruminococcaceae, whereas both of them increased in the FL group in comparison to the LPS group. Moreover, LPS increased the abundance of Bacteroidaceae and Muribaculaceae compared to the CON group, but dietary 100 mg/kg FA supplementation decreased their abundance in comparison to the LPS group.

At the genus level (Figure 10C), the ileal microbiota mainly was composed of *Bacteroides* (22.96%), *Rikenellaceae_RC9_gut_group* (9.57%), *Alistipes* (7.32%), *Lactobacillus* (7.56%), *Faecalibacterium* (4.14%), *CHKCI001* (4.28%), *Methanobrevibacter* (4.16%), *[Ruminococcus]_torques_group* (3.50%), *Barnesiella* (1.18%), and *Megamonas* (0.82%). The challenge of LPS decreased the abundance of *Lactobacillus*, *Faecalibacterium*, *CHKCI001*, and *Methanobrevibacter*, but increased the abundance of *Bacteroides*, *[Ruminococcus]_torques_group*, and *Megamonas*. However, dietary 100 mg/kg FA supplementation increased the relative abundance of *Lactobacillus*, *Faecalibacterium*, *CHKCI001*, and *Methanobrevibacter*, while decreasing the proportions of *Bacteroides*, *[Ruminococcus]_torques_group* and *Megamonas* compared to the LPS group.

## 3. Discussion

The gastrointestinal tract is an essential organ for nutrient absorption and a fundamental protective barrier against incursions of bacteria, pathogens, and toxins; therefore, maintaining intestinal homeostasis is of great importance [29]. There is increasing evidence that suggests LPS, a common toxin in the poultry industry [30], leads to loss of body weight [31], intestinal epithelium injury [32], increased intestinal permeability [33,34], and microflora dysbiosis [35]. Bioactive agents with various biological benefits [36], such as FA, may represent an interesting solution.

The intestinal epithelium is renewed from the proliferation and differentiation of multipotential stem cells located in the crypt. At the tips of the villus, fully differentiated cells are extruded into the lumen [37]. Higher VH, VH/CD, and lower CD result in a greater mucosa surface area and faster migration, which is helpful to improve the capacity of intestinal digestion and absorption [38,39]. In this study, LPS significantly impaired the intestinal morphological structure through diminishing ileal VH. This result indicates that LPS impaired intestinal digestion and absorption, which is the leading cause of decreased body weight and average daily feed intake reported in our previous study [40]. Previous studies by Gadde et al. [41] and An et al. [42] reported similar findings. It shows that FA improves growth performance by maintaining the intestinal structure of LPS-challenged broilers. However, the protective effect of FA on intestinal morphology and barrier function was reported by a previous study [43] and was possibly related to its activation of the Nrf/HO-1 signaling pathway [44], as well as its radial scavenging property [45], as evidenced by the enhanced antioxidant ability of the intestinal mucosa in this study.

Intestinal permeability is a functional indicator to reflect the integrity of the intestinal wall and the extent of bacteria translocation. The increased level of serum DAO or D-LA has been considered a token of intestinal structural damage. DAO is an intracellular enzyme that exists in intestinal mucosa [46]. Similarly, D-LA is located in the gut and is produced by various bacteria as a fermentation product. Increased serum levels of DAO and D-LA reflect the alteration of intestinal permeability as a consequence of intestinal barrier dysfunction [47]. The results in the present study showed an increased serum activity of DAO and level of D-LA in the broilers challenged with LPS, which was consistent with the reports by Yang’s study [48]. In comparison, FA-supplemented chickens decreased the activity of DAO and the levels of D-LA, indicating an augment of intestinal wall stability, which was also observed in tilapia fed with an oxidized fish oil diet [49].

Interestingly, this study found that LPS blocks ileal cell entry into the S and G_2_/M phase, leading to an aggregation of G_0_G_1_ cells and a remarkable reduction in the PI index; however, FA supplementation promoted the proliferation of the ileal cells. In the present study, LPS downregulated the expression of *CLDN-1*, *OCDN*, and *ZO-1* in the ileum. CLDN-1, OCDN, and ZO-1 belong to the tight junction proteins (TJs) and are responsible to seal the paracellular space in the epithelial cells in the gut [50]. TJs regulate ion and solute diffusion through the intercellular space and prevent the translocation of luminal antigens, microorganisms, and related toxins [51]. FA could alleviate LPS-induced TJ function loss, as evidenced by upregulated expression of *CLDN-1* and *ZO-1* in the ileum. Secretory immunoglobulin A (sIgA), the first line of defense of the gastrointestinal epithelium, is one of the most abundant antibodies in mucosal secretions. It binds to viruses and bacteria and prevents them from attaching to and invading the epithelial cells [52]. Furthermore, sIgA is reported to help with the stabilization of the microbial composition by binding commensal microbiota [53]. In line with a present study, a significant decrease in sIgA level was noticed in the LPS group [54]. Dietary FA supplementation instead increased the sIgA levels in the LPS-challenged chickens and maintained the mucosal immune response.

Intestinal microflora hemostasis provides essential health benefits to animals by promoting food digestion and nutrient absorption, influencing immune development, and providing colonization resistance against pathogens [55,56,57]. However, the beneficial effects of the intestinal microbiota depend on its composition, which have been shown to vary with a number of factors including diet, disease, and stress conditions [58,59]. In the present study, the diversity and uniformity of ileal microflora are different among the four groups. However, the composition of the ileal microflora in the FL group is highly similar to those in the CON group but differs from the LPS group, indicating that dietary FA supplementation reduces LPS-induced intestinal bacterial disturbance. Bacteroidota and Firmicutes are the two dominant phyla of bacteria in the intestine [60,61], and the F/B ratio was closely dependent on body weight because Firmicutes has higher efficiency of absorbing calorie than Bacteroidetes [62]. In this study, FA shows positive effects on the F/B ratio in the ileum of LPS-challenged chickens. Interestingly, the BW of LPS-challenged chickens was positively correlated with FA, as reported in our previous study [44]. The relative abundance of phylum Bacteroidota was found positively associated with oxidative stress [63] and was consistent with what we found in the present study, as levels of GSH and T-AOC, and the activity of SOD in the intestinal mucosa, were decreased, but the relative abundance of phylum Bacteroidetes in the ileum was increased in the LPS group, which was improved by FA. This finding is in agreement with previous observations that FA scavenges free radicals [64] and increases antioxidant function [65]. Increased relative abundance of Proteobacteria suggests a non-homeostasis of the intestinal environment [66]. In this study, chickens that receive FA supplementation showed a decrease in the relative abundance of Proteobacteria compared to the CON and LPS groups at the phylum level, which comprises several pathogens, such as Escherichia, Salmonella, and Helicobacter [67]. It is widely acknowledged that *Lactobacillus* improves intestinal health by preventing the colonization of pathogens [68]. The present results reveal that dietary FA supplementation increased the relative abundance of the family Lactobacillaceae and the genus *Lactobacillus* compared to the LPS group. *Faecalibacterium* has been recognized as a probiotic that improves epithelial proliferation and is a major butyrate producer [69]. At the genus level, FA-supplemented chickens shows an increased relative abundance of *Faecalibacterium* in the ileum compared to the ones challenged by LPS, which is in agreement with a previous study [70].

## 4. Materials and Methods

### 4.1. Reagents

Ferulic acid (C_10_H_10_O_4_, CAS number: 1135-24-6) was purchased from Chengdu Herbpurify Co. Ltd. (Chengdu, China) with 99.91% purity assessed by high-performance liquid (Welch C18 column (4.6 × 250 mm, 5 μm) with mobile phase acetonitrile: 20% phosphoric acid (22:78), Palo Alto, CA, USA). Lipopolysaccharide from *Escherichia coli* O55:B5 (LPS, L2880) was purchased from Sigma-Aldrich Corp (St. Louis, MO, USA) with ≥98% purity and ≥500,000 EU/mg titer. Then, LPS was suspended in sterile saline at a concentration of 100 μg/mL.

### 4.2. Experimental Design

Tianfu broiler chickens (*n* = 160, male, 25-day-old initial body weight 184.33 ± 3.34 g), were divided into four groups at random. Each group had four identical cages with 10 chickens per cage. Figure 11 represents the overview of the experimental design: CON group (basal diet and non-LPS challenge), LPS group (basal diet and 1.0 mg/kg LPS challenge), FA group (100 mg/kg FA + basal diet and non-LPS challenge). and FL group (100 mg/kg FA + basal diet and 1.0 mg/kg LPS challenge). On days 14, 16, 18, and 20, the birds were administered 1.0 mg/kg LPS of body weight intraperitoneally or an equal volume of normal saline based on a protocol described in a previous study [71].

All chickens were provided by and fed in the Poultry Breeding Research Unit of Sichuan Agricultural University (Ya’an, China) and were housed in an environment-controlled room with a relative humidity of 50–55%, room temperature of 22 °C, and 16 h light:8 h dark, and food and water ad libitum. The basic nutrient fact composition and of the basal diet are listed in Table 1, and the nutritional requirement was formulated according to the National Research Council requirements for chickens.

### 4.3. Sample Collection and Measurement

On day 21 of the study, eight chickens per group were selected randomly for blood sample collection. Isolated serum samples were stored at −20 °C. Intestinal samples were collected after birds were slaughtered under anesthesia. Part of the intestine was fixed in 4% (wt/vol) paraformaldehyde. Some ileal mucosa was scraped and stored at −80 °C, and another was crushed, centrifuged at 600× *g* for 5 min, and suspended at a density of 1 × 10^6^ cells/mL in the phosphate-buffered saline (PBS; Sigma-Aldrich, MO, USA) for subsequent flow cytometry analysis.

### 4.4. Morphological Analysis

Fixed intestinal samples were dehydrated using graded ethanol, and vitrificated by dimethylbenzene. Then samples were embedded in paraffin, sliced with a Lecia RM2235 microtome (Leica Biosystems Inc., Buffalo Grove, IL, USA), and stained by hematoxylin-eosin (HE). The villus height (VH) and crypt depth (CD) were measured using a microscope imaging system (DM 1000, Leica, Germany) and Image-Pro Plus software 6.0 (Media Cybernetics, Inc., Washington, DC, USA) to calculate the ratio of the villus height to the crypt depth (VH/CD).

### 4.5. Permeability and sIgA Content Analysis

The levels of diamine oxidase (DAO), d-lactate acid (D-LA) in serum, and sIgA content in ileal mucosa were determined by the chicken-specific enzyme-linked immunosorbent assay (ELISA) kits (Shanghai Enzyme-linked Biotechnology Co., Ltd., Shanghai, China).

### 4.6. Antioxidant Parameters

Biochemistry kits (Nanjing Jiancheng Bioengineering Institute, Nanjing, China) were used to measure the concentrations of superoxide dismutase (SOD), total antioxidant (T-AOC), and glutathione (GSH) in the ileal mucosa.

### 4.7. Life Cycle of Ileum Epithelium

After being fixed in cold ethanol, the cell suspension was stained with PI/Rnase Staining Buffer (BD Bioscience, Franklin Lakes, NJ, USA), resuspended in PSB, and detected by a CytoFLEX flow cytometer (Backman Coulter, Brea, CA, USA). The formula used to determine the proliferation index (PI) was (S1 + G_2_M)/(G_0/1_ + S + G_2_M).

### 4.8. Real-Time Quantificative PCR (RT-qPCR)

The isolation of total ileal RNA followed the user’s guidance of RNAiso Plus (TaKaRa Bio Inc., Shiga, Japan). The PrimeScript™ reagent Kit with gDNA Eraser Kit (TaKaRa Bio Inc., Shiga, Japan) was used for the synthesis of the first strand (cDNA). A CFX96 Real-Time PCR Detection System (Bio-Rad, Hercules, CA, USA) with a SYBR Premix Ex Taq II (TaKaRa Bio Inc., Shiga, Japan) was used for RT-qPCR and gene expression was analyzed using the 2^−∆∆Ct^ method [72]. The primers for the target genes and *β-actin* (as a housekeeping gene) are listed in Table 2.

### 4.9. 16S rRNA Sequencing and Data Analysis

Ileal digesta contents were collected randomly from six broilers per group. Total genome DNA was extracted using the SDS method. PCR reactions were carried out using a Phusion^®^ High-Fidelity PCR Master Mix (New England Biolabs, Ipswich, MA, USA). Distinct regions V3-V4 of 16S rRNA genes were amplified using a specific primer. Sequencing libraries were generated using a TruSeq^®^ DNA PCR-Free Sample Preparation Kit (Illumina, San Diego, CA, USA) and sequenced on an Illumina NovaSeq platform after quality assessment. Paired-end reads assembly used FLASH and quality control through data filtration and chimera removal. Effective tags were finally obtained. The same operational taxonomic units (OTUs) were formed from sequences that shared a minimum 97% similarity. The QIIME software (version 1.9.1; GitHub, San Francisco, CA, USA) was used to determine Alpha diversity including ACE, Chao1, Shannon, Simpson, and Beta diversity, including PCA and PCoA.

### 4.10. Statistical Analyses

All data were analyzed by SPSS 27.0 (SPSS Inc., Chicago, IL, USA) and compared significant differences among groups with one-way analysis of variance (ANOVA) and Tukey’s multiple comparison tests. Results were presented as mean ± standard error of mean (SEM). Values of *p* < 0.05 were considered significant.

## 5. Conclusions

In conclusion, LPS decreased the intestinal antioxidant ability and impaired the intestinal morphological structure, resulting in increased intestinal permeability, diminished levels of sIgA, debilitated TJ function, impaired enterocytes’ ability to proliferate, and induced gut microbiota dysbiosis in the ileum. The aforementioned detrimental effects were ameliorated by dietary FA supplementation through enhancement of antioxidant ability, augmentation of intestinal barrier integrity, improvement of the mucosal immune response, and elevated stability of ileal microflora.

## Figures and Tables

**Figure 1 molecules-28-01720-f001:**
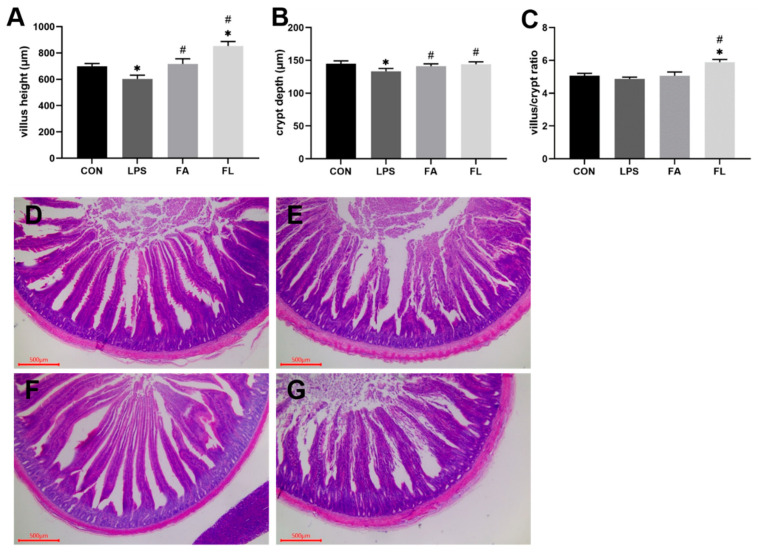
The data of villus height, crypt depth, and villus/crypt ratio. (**A**) Villus height (μm). (**B**) Crypt depth (μm). (**C**) Villus height/crypt depth ratio. Values are expressed as the mean ± SEM. *n* = 8. * means *p* < 0.05, compared with the CON group. # means *p* < 0.05, compared with the LPS group. (**D**–**G**) The representative histological changes of the ileum villus height and crypt depth with HE staining (scale bar = 500 μm). (**D**) CON group, (**E**) LPS group, (**F**) FA group, (**G**) FL group.

**Figure 2 molecules-28-01720-f002:**
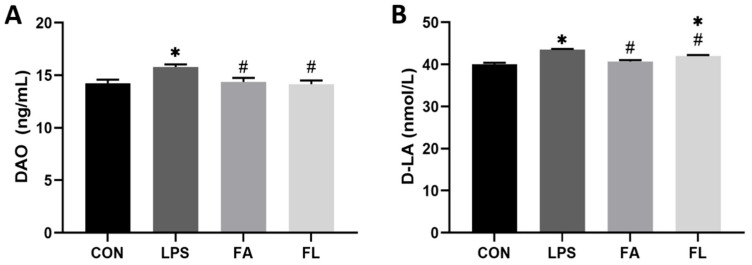
DAO activity and D-LA content in serum. (**A**) Diamine oxidase (DAO, ng/mL) (**B**) d-lactate acid (D-LA, nmol/L). Values are expressed as the mean ± SEM. *n* = 6. * means *p* < 0.05, compared with the CON group. # means *p* < 0.05, compared with the LPS group.

**Figure 3 molecules-28-01720-f003:**
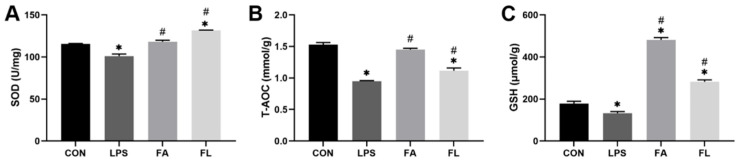
Antioxidant parameters of the intestinal mucosa. (**A**) Superoxide dismutase (SOD, U/mg), (**B**) total antioxidant capacity (T-AOC, mmol/g), (**C**) glutathione (GSH, µmol/g). Values are expressed as the mean ± SEM, *n* = 3. * means *p* < 0.05, compared with the CON group. # means *p* < 0.05, compared with the LPS group.

**Figure 4 molecules-28-01720-f004:**
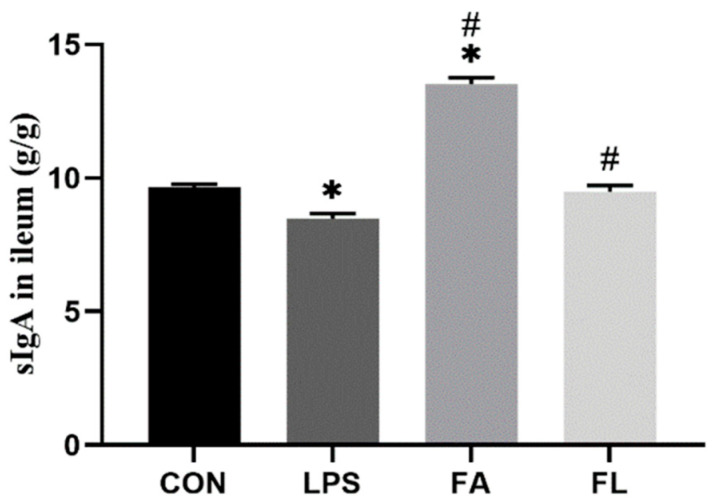
sIgA content in ileal mucosa. Values are expressed as the mean ± SEM. *n* = 4. * means *p* < 0.05, compared with the CON group. # means *p* < 0.05, compared with the LPS group.

**Figure 5 molecules-28-01720-f005:**
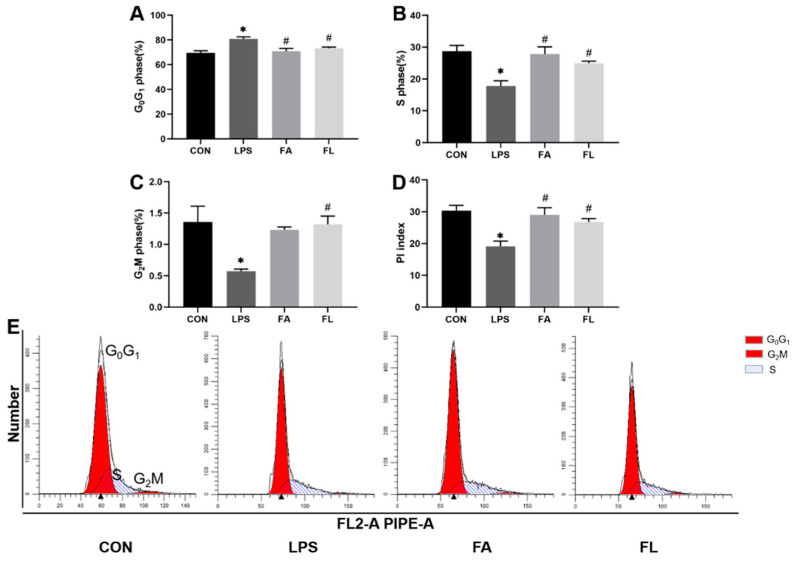
Ileal cell cycle. The percentage of the ileal cells in the (**A**) G_0_G_1_ phase (DNA pre-synthesis phase of the cell life cycle, %), (**B**) S phase (DNA synthesis phase of the cell life cycle, %), (**C**) G_2_M phase (prophase of division and division phase of the cell life cycle, %). (**D**) PI index (proliferation index). (**E**) The flow cytometry quadrant diagrams of the ileal cell life cycle among four groups. Values are expressed as the mean ± SEM. *n* = 6. * means *p* < 0.05, compared with the CON group. # means *p* < 0.05, compared with the LPS group.

**Figure 6 molecules-28-01720-f006:**
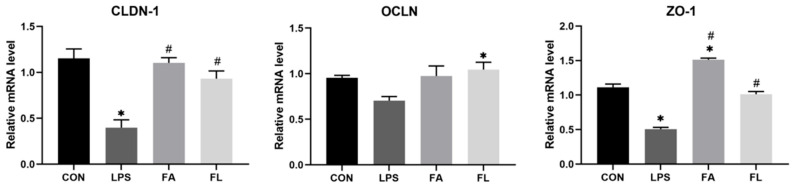
Relative mRNA expression levels of CLDN-1, OCDN, and ZO-1 in the ileum. Values are expressed as the mean ± SEM. *n* = 3. * means *p* < 0.05, compared with the CON group. # means *p* < 0.05, compared with the LPS group.

**Figure 7 molecules-28-01720-f007:**
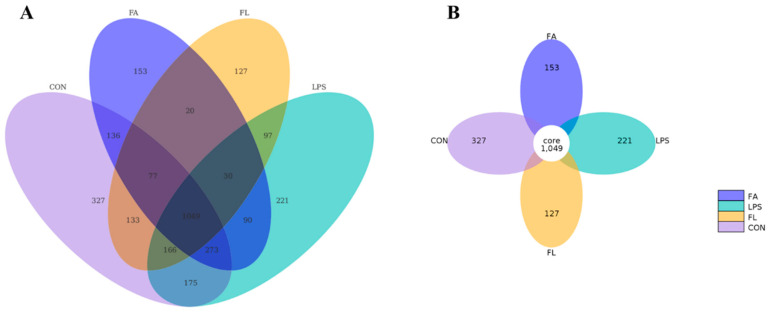
Diagram of the operational taxonomic units (OTUs) distribution of the ileal digesta. (**A**) Venn diagram: each circle represents one group. The overlapping parts represent the same number of OTUs among groups. (**B**) Flower diagram: different colors represent different groups. The core means the same number of OTUs shared among four groups. *n* = 6.

**Figure 8 molecules-28-01720-f008:**
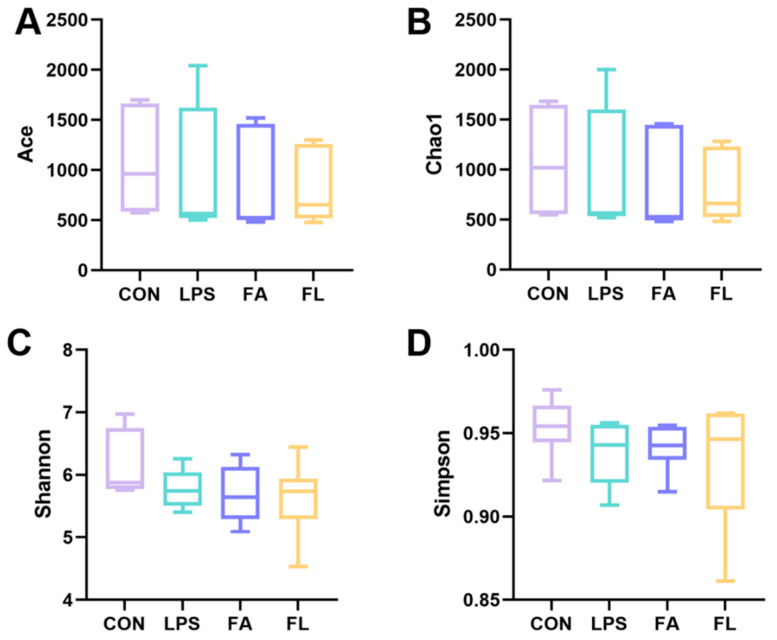
Alpha diversity of ileum microbiota. The alpha diversity was evaluated by the (**A**) Ace index, (**B**) Chao1 index, (**C**) Shannon index, and (**D**) Simpson index.

**Figure 9 molecules-28-01720-f009:**
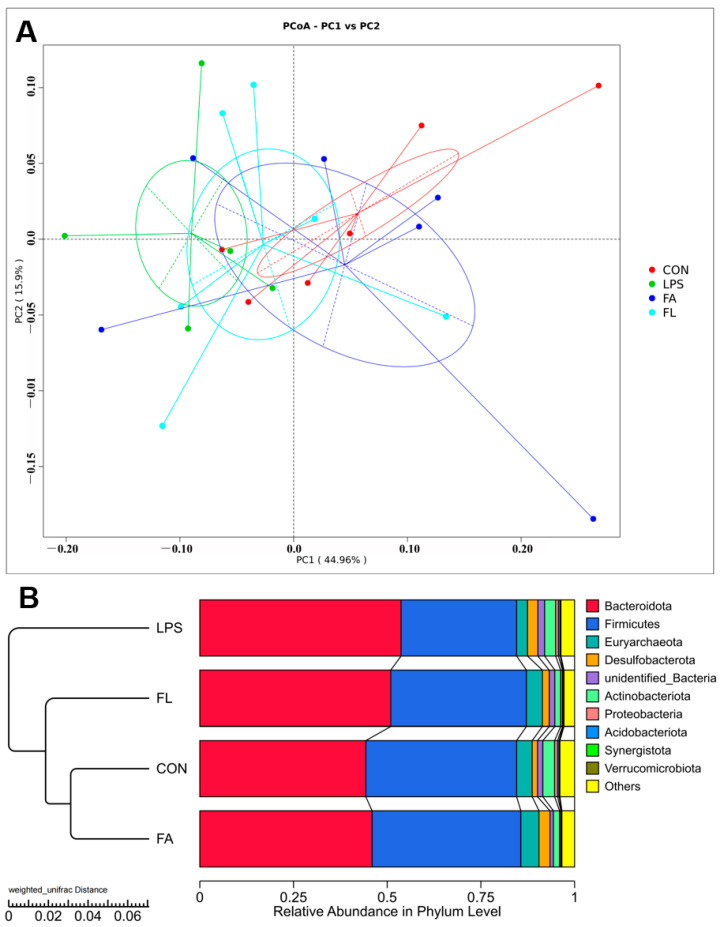
Beta diversity of ileal microbiota (PCoA). (**A**) Principal coordinate analysis based on weighted unifrac distances. (**B**) Unweighted pair group method with arithmetic mean based on weighted unifrac distances.

**Figure 10 molecules-28-01720-f010:**
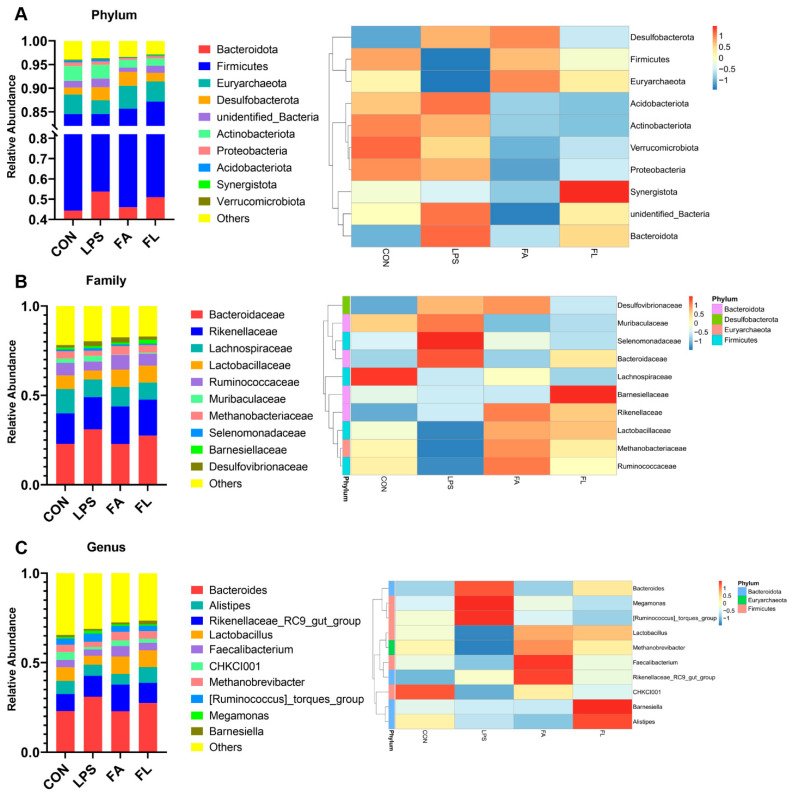
Microbial composition at the phylum, family, and genus levels in the ileum. Left: the histogram; Right: the heatmap. (**A**) Microbial structure at the phylum level. (**B**) Microbial structure at the family level. (**C**) Microbial structure at the genus level.

**Figure 11 molecules-28-01720-f011:**
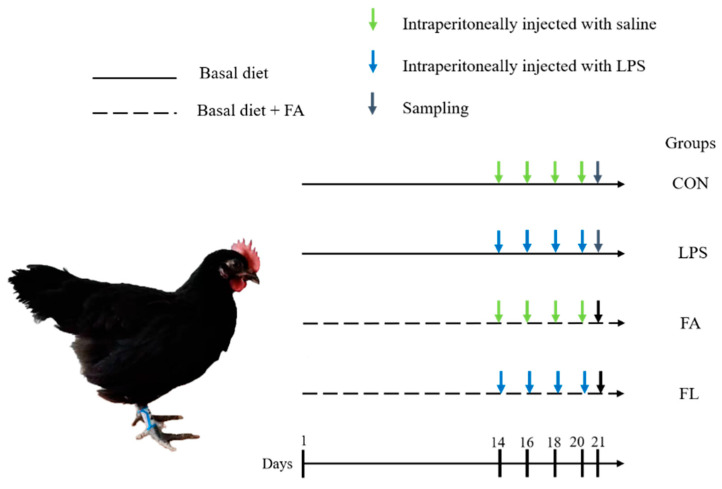
Experimental design. Four treatment groups were randomly assigned to broilers of similar body weights. CON, birds received basal diet and intraperitoneal injection of saline. LPS, birds received basal diet and intraperitoneal injection of LPS. FA, birds received basal diet supplemented with FA and intraperitoneal injection of saline. FL, birds received basal diet supplemented with FA and intraperitoneal injection with LPS.

**Table 1 molecules-28-01720-t001:** Ingredient composition and nutrient content of the basal diet.

Ingredient	%	Calculated Nutrients	%
Corn	59.50	Metabolizable energy (MJ kg^−1^)	12.80
Soybean meal	32.90	Crude protein	19.70
Vegetable oil	4.65	Lysine	1.08
CaCO_3_	0.50	Methionine	0.40
CaHPO_4_	1.60	Methionine + Cystine	0.74
NaCl	0.30	Calcium	0.77
Choline	0.10	Nonphytate P	0.40
DL-Met	0.12		
Premix ^1^	0.33		
Total	100		

^1^ Provided per kg for diet: vitamin A (all-trans retinol acetate), 12,500 IU; cholecalciferol, 2500 IU; vitamin E (all-rac-a-tocopherol acetate), 18.75 IU; vitamin K (menadione Na bisulfate), 5.0 mg; thiamine (thiamine mononitrate), 2.5 mg; riboflavin, 7.5 mg; vitamin B6, 5.0 mg; vitamin B12, 0.0025 mg; pantothenate, 15 mg; niacin, 50 mg; folic acid, 1.25 mg; biotin, 0.12 mg; Cu (CuSO_4_·5H_2_O), 10 mg; Mn (MnSO_4_·H_2_O), 100 mg; Zn (ZnSO_4_·7H_2_O), 100 mg; Fe (FeSO_4_·7H_2_O), 100 mg; I (KI), 0.4 mg; and Se (Na_2_SeO_3_), 0.2 mg.

**Table 2 molecules-28-01720-t002:** Primers used for qRT-PCR.

Gene	Accession Number	Primer Sequence (5′–3′)	Product Length (bp)
*CLDN-1*	XM_001013611.2	F: CATACTCCTGGGTCTGGTTGGT R: GACAGCCATCCGCATCTTCT	100
*OCLN*	NM_205128.1	F: CTCAATCAGCTCAGCCGAC R: TCTCCTGCTTCTTGCTTTGGTA	130
*ZO-1*	NM_040706827.1	F: GTAAACCACTGCCTACACC R: ATATCTTAACTCTACTTCGCACA	90
*β-actin*	NM_205518.1	F: AAGGATCTGTATGCCAACACA R: AGACAGAGTACTTGCGCTCA	148

*CLDN-1*: Claudin-1, *OCLN*: Occludin, *ZO-1*: zonula occkudens-1, *β-actin*: reference gene.

## Data Availability

The raw data supporting the conclusions of this article will be made available by the authors, without reservation.

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
