# Peer review of "Effects of Dietary Ferulic Acid on Intestinal Health and Ileal Microbiota of Tianfu Broilers Challenged with Lipopolysaccharide"

_molecules, 2023, doi:10.3390/molecules28041720_

Round 1

Reviewer 1 Report

This is a comprehensive study encompassed required analysis with enough replicates. In my opinion this paper is suitable for publication in Molecules after revision required revision. 

The word “ameliorate” use is too many. Find alternative words to improve readability

L65-67: check the sentence

Table 2: Product length of CLDN-1 should be 100 bp not 82.

https://www.ncbi.nlm.nih.gov/tools/primer-blast/primertool.cgi?ctg_time=1674462712&job_key=6uA0wxunFg8xNYYwi1CiAvFLszDcWKgt3Q

L348: İleal digesta content instead ileal fecal content

L347: 16S rRNA Sequencing and Data Analysis part is too simple. Please give more details.

L354: Two-way anova would be more appropriate rather than one way anova. The model should contain diet (basal vs FA) and LPS (challenged vs non-challenged) as main factors and two way interactions. Result section, graphs and discussion should be updated accordingly.

Please explain how did you determine 100mg/kg FA level in diet?

L241-L242: by “promoting” nutrient absoption.

L245: including diet, disease and “stress conditions”

L251: absorption of what?

L261-263: suggestion: In this study, chickens receive FA supplementation showed decrease in relative abundance of Proteobacteria compared to the CON and LPS groups at phylum level, which comprises several pathogens, such as Escherichia, Salmonella, and Helicobacter [67].

L267: as “a probiotic”

L267 and L269: Faecalibacterium instead “Faecalibatterium”

L267-268: “improves”

Author Response

Response to Reviewer 1 Comments

Point 1: The word “ameliorate” use is too many. Find alternative words to improve readability

Response 1: In line 25, changed the “ameliorated” to “improved”.

In line 29, changed the “ameliorated” to “rectified”.

In line 30, changed “ameliorate” to “decreased”.

In line 61, changed “ameliorates” to “reduces”.

In line 62, changed “amelioration in” to “regulating”.

In line 94, changed “ameliorated” to “modified”.

In line 258, changed “ameliorated” to “improved”.

Point 2: L65-67: check the sentence

Response 2: No changes.

Point 3: Table 2: Product length of CLDN-1 should be 100 bp not 82.

https://www.ncbi.nlm.nih.gov/tools/primerblast/primertool.cgi?ctg_time=1674462712&job_key=6uA0wxunFg8xNYYwi1CiAvFLszDcWKgt3Q

Response 3: changed to 100

Point 4: L348: Ileal digesta content instead ileal fecal content

Response 4: Changed “Ileal digesta content” to “Ileal fecal content”

Point 5: L347: 16S rRNA Sequencing and Data Analysis part is too simple. Please give more details.

Response 5: Added more details as follows: Total genome DNA was extracted using SDS method. PCR reactions were carried out of Phusion® High-Fidelity PCR Master Mix (New England Biolabs, New Jersey, USA). Distinct regions V3-V4 of 16S rRNA genes were amplified using specific primer. Se-quencing libraries were generated using TruSeq® DNA PCR-Free Sample Preparation Kit (Illumina, San Diego, USA) and sequenced on an Illumina NovaSeq platform after quality assessment. Paired-end reads assembly using FLASH and quality control through data filtration and chimera removal. Effective tags finally obtained. The same operational taxonomic units (OTUs) were formed from sequences that shared a mini-mum 97% similarity. The QIIME software (version 1.9.1; GitHub, San Francisco, CA) was used to determine Alpha diversity including ACE, Chao1, Shannon, Simpson and Beta diversity including PCA and PCoA.

Point 6: L354: Two-way anova would be more appropriate rather than one way anova. The model should contain diet (basal vs FA) and LPS (challenged vs non-challenged) as main factors and two way interactions. Result section, graphs and discussion should be updated accordingly.

Response 6: The study aimed to explore the positive effects of Ferulic acid on intestine of chicken challenged with LPS. Compare the LPS group with CON group to show the negative effects of LPS or provide a certify that chicken in LPS group have intestinal injury. At the same, compare the FL group with LPS group to show whether Ferulic acid has therapeutic effect and compare the FL group with CON group to show how the therapeutic effect of Ferulic acid on chicken challenged with LPS. It is not necessary to explore the interaction between LPS and Ferulic acid on chicken. Therefor, the study used one-way anova.

Point 7: Please explain how did you determine 100mg/kg FA level in diet?

Response 7: In fact, information regards to protective effects of FA on liver and intestinal damage in LPS-induced broilers remains unreported. At present, most of the study mainly explore the effects of FA in rats, mice and a few in piglets and zebrafish. In addition, the dosage and mode of administration are different. It is hard to make out what the standard to use. We can not find the effective information about the dosage of FA to be used in chickens. So, we determined the dosage through a simple pre-experiment based on the number of loose stools and liver histopathology.

Point 8: L241-L242: by “promoting” nutrient absoption.

Response 8: changed to “promoting food digestion and nutrient absorption”.

Point 9: L245: including diet, disease and “stress conditions”

Response 9: add “stress conditions”.

Point 10: L251: absorption of what?

Response 10: changed to “higher efficiency of absorbing calorie”

Point 11: L261-263: suggestion: In this study, chickens receive FA supplementation showed decrease in relative abundance of Proteobacteria compared to the CON and LPS groups at phylum level, which comprises several pathogens, such as Escherichia, Salmonella, and Helicobacter [67].

Response 11: changed “phylum proteobacteria compared to the CON and LPS groups” to “proteobacteria compared to the CON and LPS groups at phylum level”.

Point 12: L267: as “a probiotic”

Response 12: changed “aprobiotic” to “a probiotic”

Point 13: L267 and L269: Faecalibacterium instead “Faecalibatterium”

Response 13: changed “Faecalibatterium” to “Faecalibacterium”.

Point 14: L267-268: “improves”

Response 14: changed to improves.

Reviewer 2 Report

Ferulic acid (FA) is one of the active plant components and is mainly found in several Chinese medical plants. Generally, it has been recognized as an effective antioxidant, but some previous studies reveals that it possesses a variety of biological activities in body. The experimental design in the study is reasonable, with readable writing, and the category meets the journal scope. In this study, the FA improves the intestinal morphological structure and permeability by increasing the TJs expression, antioxidant capacity, and sIgA content in the intestinal mucosa and modulating the ileal microbial composition in broilers. The results answer the questions proposed by the initial hypothesis, besides, from the aspects of intestinal permeability, histomorphology, biochemistry parameters, and microbial composition exhibited how the FA affected the intestine health of the Tianfu broiler. The results of this study provide us more practical information for the real application of FA in the feed industry of broiler. In my opinion, the research work is meaningful and deserves to be accepted. However, there are still a series of important problems that need to be improved, after which the paper will be more perfect or reasonable. In particular, the information obtained by the readers will be more comprehensive. In the end, the proposal for this paper is a major revision and the correct suggestions are listed as below.

1.     The section “experimental animals and management” (4.2, line 279) and the section “study design” (4.3, line 293) can be combined into one section, and this will be more convenient for readers to find out the key information quickly.

2.     In the study, each diet group had four cages with 10 chickens per cage (line 295), while on day 21, eight chickens per diet group were selected for sampling (line 309), and six broilers per group were collected for the 16s rRNA analysis (line 348). So, the question is how did you select the eight chickens for slaughtering, randomly or based on the medium body weight? Describing it in detail, please. Besides, exception for the initial body weight of broilers, please provide the information about the initial day of age for the chickens, it is important for the other researchers. In addition, why did you not prepare eight cages with 10 chickens per cage before? if so, I suppose the sampling protocol would be more scientific.

3.     In line 280, wrong verb tense, the “feed” replaced with “fed”.

4.     In line 285, supplementation of the publication time of the NRC for chicken.

5.     In line 286 (table 1), in the table, the first line addition of the “Ingredient”, then addition of the “total” and total value of the proportion at the bottom of the part “Ingredient”. Besides, based on the nutritional requirements of NRC (1994) for the chicken in growth stage (0-8 weeks), the level of calcium and ME recommended is 0.9-1.0 and 13.39 MJ/kg, respectively. So, whether the present content of calcium (0.77) and ME (12.80) in the table is sufficient to meet the real growth requirement of the chickens?

6.     In the part 4.3 (line 298), the dietary dosage of FA is 100 mg/kg, why did you chose this level? Unfortunately, we haven’t found any information related to it in the part Introduction (line 37). Providing the relative information in this part, please.

7.     In part 4.6 (line 320), providing the relevant CAS number information for these commercial kits.

8.     In part 4.11 (line 354), there several problems in describing the statistical analysis. The one-way ANOVA is one of the procedures attached in the statistical software SPSS, and the significance among diet groups is determined based on the multiple comparison, please supply the information which multiple comparison you used before. In addition, under the figure legend in the whole context, we haven’t found any information about the effective statistical number (replication n). As you know, it is one of the important information that helps the readers to well understand the data you analyzed, supplementation of the information in the part, please.

9.     In line 267, the “aprobiotic” replaced with the “aerobiotic”.

10.  In line 170, changing the “FI” to “FL”.

11.  In line 163, changing the “Figure 11” to “Figure 10”.

12.  In line 159 (Figure 9), for figure 9A, the principal coordinate analysis was based on the Weighted UniFrac distance, according to the microbial data in the text, we can easy to judge that the relative abundance of various ileal species is different. Hence, we suggest that the PCA should be analyzed based on the Weighted UniFrac distance. So, the figure 9B is not suitable placed in here. In addition, several numbers in the figures are not clear enough to read, splitting or modulating the graph again, please.

13.  In line 32, “elevates” is not suitable to describe the microbial composition, maybe the verbs modulate or change will be more reasonable.

14.  In the whole context, the microbial names are not in italics except at the genus or species levels, please correct it.

15.  For the figure 10 (line 188), it is very good to use the heatmap and the histogram together to show the microbial composition, if the heatmaps are all marked with the “*” or other symbols to demonstrate the significantly difference in the relative abundance of microorganisms among diet groups the figures will be more perfect. Hence, the statistical analysis of the microbial relative abundance is necessary.

16.  Usually, the diet formula is placed together with the results of growth performance of broilers, how did you consider it before? If you place the diet formula in this paper, it is better to submit it as supplemental material.

17.  For the part 2.3 (line 88), considering you were not present the performance data of the broilers in this paper, the results of intestinal morphology present in first will be more reasonable, the other parts keep the original order.

Author Response

Response to Reviewer 3 Comments

Point 1: The section “experimental animals and management” (4.2, line 279) and the section “study design” (4.3, line 293) can be combined into one section, and this will be more convenient for readers to find out the key information quickly.

Response 1: We had revised this part according to your suggestions in paper.

Point 2: In the study, each diet group had four cages with 10 chickens per cage (line 295), while on day 21, eight chickens per diet group were selected for sampling (line 309), and six broilers per group were collected for the 16s rRNA analysis (line 348). So, the question is how did you select the eight chickens for slaughtering, randomly or based on the medium body weight? Describing it in detail, please. Besides, exception for the initial body weight of broilers, please provide the information about the initial day of age for the chickens, it is important for the other researchers. In addition, why did you not prepare eight cages with 10 chickens per cage before? if so, I suppose the sampling protocol would be more scientific.

Response 2: We chose eight chickens for slaughtering randomly.

The initial day of age is 25 days and we have added the information in article.

Actually, the room was not big enough to hold 32 cages and each cage could accommodate 10 chickens.

Point 3: In line 280, wrong verb tense, the “feed” replaced with “fed”.

Response 3: Instead “fed” of “feed”.

Point 4: In line 285, supplementation of the publication time of the NRC for chicken.

Response 4:1994

Point 5: In line 286 (table 1), in the table, the first line addition of the “Ingredient”, then addition of the “total” and total value of the proportion at the bottom of the part “Ingredient”. Besides, based on the nutritional requirements of NRC (1994) for the chicken in growth stage (0-8 weeks), the level of calcium and ME recommended is 0.9-1.0 and 13.39 MJ/kg, respectively. So, whether the present content of calcium (0.77) and ME (12.80) in the table is sufficient to meet the real growth requirement of the chickens?

Response 5:Tianfu broiler is one kind of Chinese native chicken breeds and mainly distributed in Southwest China. Tianfu broiler grows more slowly than white feather broiler and the nutritional requirement is not as high.  

Point 6: In the part 4.3 (line 298), the dietary dosage of FA is 100 mg/kg, why did you chose this level? Unfortunately, we haven’t found any information related to it in the part Introduction (line 37). Providing the relative information in this part, please.

Response 6:In fact, information regards to protective effects of FA on liver and intestinal damage in LPS-induced broilers remains unreported. At present, most of the study mainly explore the effects of FA in rats, mice and a few in piglets and zebrafish. In addition, the dosage and mode of administration are different. It is hard to make out what the standard to use. We can’t find the effective information about the dosage of FA to be used in chickens. So, we determined the dosage through a simple pre-experiment based on the number of loose stools and liver histopathology.

Point 7: In part 4.6 (line 320), providing the relevant CAS number information for these commercial kits.

Response 7:The commercial kits have the article number instead of the CAS number.

Point 8: In part 4.11 (line 354), there several problems in describing the statistical analysis. The one-way ANOVA is one of the procedures attached in the statistical software SPSS, and the significance among diet groups is determined based on the multiple comparison, please supply the information which multiple comparison you used before. In addition, under the figure legend in the whole context, we haven’t found any information about the effective statistical number (replication n). As you know, it is one of the important information that helps the readers to well understand the data you analyzed, supplementation of the information in the part, please.

Response 8: We used Tukey’s multiple comparisons test and added the information about the effective statistical number.

Point 9: In line 267, the “aprobiotic” replaced with the “aerobiotic”.

Response 9: In fact, it is a probiotic.

Point 10: In line 170, changing the “FI” to “FL”.

Response 10: We corrected “FI” to “FL”.

Point 11: In line 163, changing the “Figure 11” to “Figure 10”.

Response 11: We had corrected “Figure 11” to “Figure 10”.

Point 12: In line 159 (Figure 9), for figure 9A, the principal coordinate analysis was based on the Weighted UniFrac distance, according to the microbial data in the text, we can easy to judge that the relative abundance of various ileal species is different. Hence, we suggest that the PCA should be analyzed based on the Weighted UniFrac distance. So, the figure 9B is not suitable placed in here. In addition, several numbers in the figures are not clear enough to read, splitting or modulating the graph again, please.

Response 12: We deleted the figure 9B and provided the higher quality figures.

Point 13: In line 32, “elevates” is not suitable to describe the microbial composition, maybe the verbs modulate or change will be more reasonable.

Response 13: We changed “elevates secretory immunoglobulin A (sIgA) level and ileal microflora composition” to “elevates secretory immunoglobulin A (sIgA) level and modulates ileal microflora composition”.

Point 14: In the whole context, the microbial names are not in italics except at the genus or species levels, please correct it.

Response 14: We have corrected the microbial names to be not-italic in the phylum and family levels.

Point 15: For the figure 10 (line 188), it is very good to use the heatmap and the histogram together to show the microbial composition, if the heatmaps are all marked with the “*” or other symbols to demonstrate the significantly difference in the relative abundance of microorganisms among diet groups the figures will be more perfect. Hence, the statistical analysis of the microbial relative abundance is necessary.

Response 15: There is no significant difference.

Point 16: Usually, the diet formula is placed together with the results of growth performance of broilers, how did you consider it before? If you place the diet formula in this paper, it is better to submit it as supplemental material.

Response 16: The growth performance data had been published in another paper.

https://www.mdpi.com/2072-6651/14/3/227

Point 17: For the part 2.3 (line 88), considering you were not present the performance data of the broilers in this paper, the results of intestinal morphology present in first will be more reasonable, the other parts keep the original order.

Response 17: We had adjusted the order of the part of intestinal morphology according to your suggestion.

Round 2

Reviewer 1 Report

Table 1: Should be "Methionine" not Methionime

Author Response

Point 1: Table 1: Should be "Methionine" not Methionime.

Response: Corrected to “Methionine”.

Reviewer 2 Report

Point 6: In the part 4.3 (line 298), the dietary dosage of FA is 100 mg/kg, why did you chose this level? Unfortunately, we haven’t found any information related to it in the part Introduction (line 37). Providing the relative information in this part, please.

Response 6In fact, information regards to protective effects of FA on liver and intestinal damage in LPS-induced broilers remains unreported. At present, most of the study mainly explore the effects of FA in rats, mice and a few in piglets and zebrafish. In addition, the dosage and mode of administration are different. It is hard to make out what the standard to use. We can’t find the effective information about the dosage of FA to be used in chickens. So, we determined the dosage through a simple pre-experiment based on the number of loose stools and liver histopathology.

Advice: This reason should be briefly declared in the part Introduction.

Point 16: Usually, the diet formula is placed together with the results of growth performance of broilers, how did you consider it before? If you place the diet formula in this paper, it is better to submit it as supplemental material.

Response 16: The growth performance data had been published in another paper.

https://www.mdpi.com/2072-6651/14/3/227

Author Response

Point 6: In the part 4.3 (line 298), the dietary dosage of FA is 100 mg/kg, why did you chose this level? Unfortunately, we haven’t found any information related to it in the part Introduction (line 37). Providing the relative information in this part, please.

Response 6:In fact, information regards to protective effects of FA on liver and intestinal damage in LPS-induced broilers remains unreported. At present, most of the study mainly explore the effects of FA in rats, mice and a few in piglets and zebrafish. In addition, the dosage and mode of administration are different. It is hard to make out what the standard to use. We can’t find the effective information about the dosage of FA to be used in chickens. So, we determined the dosage through a simple pre-experiment based on the number of loose stools and liver histopathology.

Advice: This reason should be briefly declared in the part Introduction.

Response: In this study, we do not to explore the best dosage of FA in broilers or whether the interventional effect is in a dose-dependent manner. However, the purpose is to explore whether FA has positive effects on LPS-challenged Tianfu broilers. In the part Introduction, we just declared that the relevant study remained unreported. (In line 62-64).

Point 16: Usually, the diet formula is placed together with the results of growth performance of broilers, how did you consider it before? If you place the diet formula in this paper, it is better to submit it as supplemental material.

Response 16: The growth performance data had been published in another paper.

https://www.mdpi.com/2072-6651/14/3/227

Response: In line 206-208, we cited the paper containing growth performance data to illustrate that ferulic acid ameliorates intestinal structure and thus results in increased growth performance of broilers with LPS challenge.